# Gene Expression Profiling of NFATc1-Knockdown in RAW 264.7 Cells: An Alternative Pathway for Macrophage Differentiation

**DOI:** 10.3390/cells8020131

**Published:** 2019-02-07

**Authors:** Roberta Russo, Selene Mallia, Francesca Zito, Nadia Lampiasi

**Affiliations:** Institute of Biomedicine and Molecular Immunology “Alberto Monroy”, National Research Council, Via Ugo La Malfa 153, 90146 Palermo, Italy; roberta.russo@ibim.cnr.it (R.R.); selene.mallia93@gmail.com (S.M.); zito@ibim.cnr.it (F.Z.)

**Keywords:** NFATc1-knockdown, GATA2, STAT6, transcription factors, macrophages, osteoclasts

## Abstract

NFATc1, which is ubiquitous in many cell types, is the master regulator of osteoclastogenesis. However, the molecular mechanisms by which NFATc1 drives its transcriptional program to produce osteoclasts from macrophages (M) remains poorly understood. We performed quantitative PCR (QPCR) arrays and bioinformatic analyses to discover new direct and indirect NFATc1 targets. The results revealed that NFATc1 significantly modified the expression of 55 genes in untransfected cells and 31 genes after NFATc1-knockdown (≥2). Among them, we focused on 19 common genes that showed changes in the PCR arrays between the two groups of cells. Gene Ontology (GO) demonstrated that genes related to cell differentiation and the development process were significantly (*p* > 0.05) affected by NFATc1-knockdown. Among all the genes analyzed, we focused on GATA2, which was up-regulated in NFATc1-knockdown cells, while its expression was reduced after NFATc1 rescue. Thus, we suggest GATA2 as a new target of NFATc1. Ingenuity Pathway Analysis (IPA) identified up-regulated GATA2 and the STAT family members as principal nodes involved in cell differentiation. Mechanistically, we demonstrated that STAT6 was activated in parallel with GATA2 in NFATc1-knockdown cells. We suggest an alternative pathway for macrophage differentiation in the absence of NFATc1 due to the GATA2 transcription factor.

## 1. Introduction

It is now well established that pluripotency of the cells is maintained as a consequence of the balance of different lineage-specifying forces. Instead, lineage specification is due to modifications in the combination of specific transcription factors and miRNAs that carry out a particular transcription program.

Differentiation of hematopoietic progenitors into distinct lineages depends on the activation of specific and tightly regulated transcription factors. First specification is between myeloid and erythroid lineages and depends on several factors playing critical roles. As an example, PU1 is an essential factor for reconstitution of the myeloid lineage [1], and Cebp-a, -b, and -ε factors play major roles in commitment toward myeloid cells: granulocytes, monocytes and macrophages [2,3,4]. Macrophages are heterogeneous and versatile cells that can acquire different functional phenotypes depending on the microenvironment and the resident cytokines. Two main subsets of macrophages are known, the classically activated or inflammatory (M1 type), and the alternatively activated anti-inflammatory (M2 type). They polarize in the presence of granulocyte macrophage (GM) or macrophage (M) colony-stimulating factor. On the other hand, osteoclasts derive from the commitment of hematopoietic stem cells (HSCs) into the monocyte/macrophage lineage through PU1 [5]. Afterwards, macrophages develop into pre-osteoclasts, undergoing a complex differentiation process implying a series of steps—proliferation, migration and fusion—that lead to differentiated osteoclasts (i.e., multinucleated and polarized cells with bone resorptive activity) [5]. Osteoclastogenesis depends on two main cytokines, the M-CSF and the receptor activator of nuclear factor-κB ligand (RANKL), which are essential for osteoclast lifespan and function through their receptors, the colony-stimulating factor receptor (c-fms) and receptor activator of nuclear factor-κB (RANK), respectively [6,7,8]. Therefore, osteoclasts share with macrophages a trait of differentiation due to the activation of many signalling pathways and transcription factors as a consequence of the same cytokine stimulation (M-CSF). Osteoclast differentiation is a complex mechanism that involves multiple factors such as cytokines, transcription factors and miRNAs [9]. NFATc1 is considered the master transcription factor regulating osteoclast differentiation [10] and it is maintained in the cytosol as inactive hyper-phosphorylated protein in pre-osteoclasts. During the following differentiation, calcium signalling, in particular the calcium oscillations observed in osteoclast precursors, stimulates the phosphatase calcineurin, which dephosphorylates NFATc1 and allows its nuclear translocation [10,11]. In the nucleus, NFATc1 regulates the expression of many osteoclast-specific genes such as tartrate-resistant acid phosphatase (TRAP), c-Fos, cathepsin K (CtsK) and microphthalmia transcription factor (MITF) [8].

NFATc1 is widely expressed in different cell types and is essential for the development of many tissues [12]. Although it is considered the master regulator of osteoclastogenesis, it is not clear how the ubiquitous NFATc1 can direct an osteoclast-specific transcription program. 

In this study, we tried to unravel the molecular pathway underlying pre-osteoclast differentiation by PCR array analysis, using RAW 264.7 macrophage cell line. We studied the expression profiles of untransfected and siRNA-NFATc1 transfected cells during the early steps of osteoclast differentiation after RANKL induction. Gene Ontology (GO) and Ingenuity Pathway Analysis (IPA) mechanistic analyses were performed taking into consideration the function of mRNAs, which changed their expression significantly in PCR arrays. Altogether, our results suggest that GATA2 was a new target downstream of NFATc1, whose depletion directly or indirectly caused GATA2 activation. Thus, GATA2 might be involved in the differentiation of bone marrow-derived macrophages in the absence of NFATc1, while precisely NFATc1 could downregulate it during commitment of macrophages towards the osteoclast lineage. 

Overall, we hypothesize that these two transcription factors could be involved in the specification of different cell lineages.

## 2. Materials and Methods

### 2.1. Cell Culture and Osteoclastogenesis In Vitro

The murine RAW 264.7 macrophage cell line was purchased from the American Type Culture Collection (Manassas, VA, USA). Cells were grown in Dulbecco’s Modified Eagle’s Medium (DMEM, Gibco, NY, USA) with 10% heat-inactivated fetal bovine serum (FBS, Sigma Chemical Co., St. Louis, MO, USA), 100 U/ml penicillin and 100 µg/ml streptomycin. To induce osteoclast differentiation, cells were suspended in alpha-minimal essential medium (α-MEM Gibco, NY, USA) with 10% heat-inactivated fetal bovine serum (FBS, Sigma), 100 U/ml penicillin and 100 µg/ml streptomycin with RANKL 50 ng/mL (Peprotech, USA). To identify multinucleated osteoclasts, RAW 264.7 cells (untransfected and transfected) were grown on coverslips for 24 h with or without RANKL (50 ng/mL). Cells were fixed with 4% paraformaldehyde (Sigma Chemical Co., St. Louis, MO, USA) in physiological phosphate buffered saline (PBS) for 15 min. All the coverslips were mounted on glass slides with mounting medium containing DAPI (Invitrogen, Thermo Fisher Scientific Carlsbad, CA, USA), and observed under an Axioskop-2 Plus microscope (Zeiss, Germany), equipped for epifluorescence. Images were recorded using a digital camera system, cropped and grouped, and contrast and brightness were adjusted using Photoshop CS2 (Adobe Systems, San Jose, CA, USA).

### 2.2. Small Interfering RNA (siRNA) Transfection

Cells were transfected with NFATc1-siRNA or non-correlated (NC) siRNA (Qiagen, Germantown, Maryland, USA) as previously reported [13]. In brief, cells (2.5 × 10^5^) were seeded onto 6-well plates in medium without antibiotics; 24 h later, the transfection of siRNAs was carried out with Lipofectamine RNAiMAX (Invitrogen, Thermo Fisher Scientific Carlsbad, CA, USA). All transfections were carried out with 20 μM duplex siRNA in medium without FBS or antibiotics. Then, 6 h later, we added RANKL (50 ng/mL) to the medium. One or two days after transfection, cells were recovered to perform further analyses. Experiments were repeated three times.

### 2.3. RNA Extraction and cDNA Synthesis

RAW 264.7 cells were cultured (1 × 10^6^ cells/well) in a 6-well plate overnight. Cells were untransfected or transfected with specific siRNA and treated with RANKL 50 ng/mL for 24, 48 and 72 h. After this time of stimulation, the cells were washed once with PBS. Total RNA was isolated using the GenElute Mammalian Total RNA Miniprep Kit (Sigma Chemical Co., St. Louis, MO, USA) and quantified by using a spectrophotometer (Eppendorf S.r.l., Hamburg, Germany). Total RNA (3μg) was reverse-transcribed to cDNA using the SuperScript Vilo (Invitrogen, Thermo Fisher Scientific Carlsbad, CA, USA). 

### 2.4. Quantitative Polymerase Chain Reaction (QPCR)

Quantification of gene expression was performed using the StepOne Plus Real-Time PCR as described in the manufacturer’s manual (Applied Biosystems Life Technologies, Carlsbad, CA, USA) with SYBR Green chemistry and the comparative threshold cycle method [14]. The quantitative PCR (QPCR) was run as follows: 1 × cycle, denaturing at 95 °C for 10 min for DNA polymerase activation; and 38 cycles, melting at 95 °C for 15 s and annealing/extension at 60 °C for 60 s. QPCR was then performed in triplicate on each cDNA sample for each gene using primers by Qiagen: QT001676692 (*NFATc1*), QT00166663 (*TRAF6*), QT00108815 (*MMP9*), QT00131313 (*MITF1*), QT00131012 (*TRAP*), QT01047032 (*DC-STAMP*), QT02589489 (*CTSK 2*), QT00149415 (*RelA*), QT00197568 (*RhoA*), QT01167257 (*OSCAR*), QT00096194 (*Myc*), QT00102193 (*Runx*2), QT00159768 (*SMAD*5), QT00103005 (*STAT4*), QT00170975 (*Hnf1*), QT00249781 (*Esr1*), QT00265846 (*Egr1*), QT00160524 (*GATA2*) and AQT01658692 (*GAPDH*). For *Fos* we used the following primers, after validation F: 5′CACTCCAAGCGGAGACAGAT3′ and R: 5′TCGGTGGGCTGCCAAAATAA3′. The threshold cycle (CT) values were calculated against the housekeeping gene *GAPDH*. At least three distinct biological samples were examined for each gene and treatment (each performed in triplicate). 

### 2.5. RT^2^ Profiler PCR Array Analysis

Mouse RT^2^ Profiler PCR array for transcription factors (PAMM-075ZC) and for osteoporosis (PAMM-170ZC) in 96-well plate format (Qiagen Sciences, Germantown, MD, USA) were used to analyze gene expression changes. Samples were prepared from pooled RNA extracted from RAW264.7 cells using the RNeasy MinElute Cleanup Kit (Qiagen Sciences, Germantown, MD, USA). RAW 264.7 cells were untransfected (+/- RANKL) or transfected +RANKL (siRNA-control/siRNA-NFATc1). Total RNA was reverse-transcribed in cDNA according to Qiagen’s instructions for the RT^2^ First Strand Kit. QPCR was run as described above, with RT^2^ SYBR Green ROX QPCR Mastermix, in a StepOne Real-Time instrument (Applied Biosystem Life Technologies, Carlsbad, CA, USA). Normalization was done on automatic HKG panel. This method uses the five genes (wells H1–H5) that have small changes in threshold cycle (CT) values across all of the included samples. Array analysis of the results was performed using the software provided by SABiosciences (www.SABiosciences.com).

### 2.6. Western Blot

RAW 264.7 cells were transfected or untransfected and cultured as described above for 24 and 72 h with RANKL. Then, cells were lysed and total cellular protein were extracted using RIPA buffer (Cell Signaling Inc. Beverly, MA, USA). The protein concentration of cell lysates was determined by the Bradford method. For western blot, 30 μg of protein was separated on 10% SDS–polyacrylamide gels by electrophoresis and transferred to a nitrocellulose membrane (Millipore Temecula, CA, USA). Membranes were incubated overnight at 4 °C with the following antibodies: NFATc1 and NFATc2 (Santa Cruz, CA, USA) at 1:1000 dilution; DC-STAMP (Millipore Temecula, CA, USA) at 1:500 dilution; GATA2 (Cloud Clone Corporation USA) at 1:500 dilution; TRAF6 (Santa Cruz, CA, USA) at 1:500 dilution; RANK (Santa Cruz, CA, USA) at 1:500 dilution; MITF (Abcam, England) at 1:500 dilution; and STAT6 (Santa Cruz, CA, USA) at 1:500 dilution. 

The secondary antibodies Alexa Fluor 680 Goat anti-Rabbit (1:2000) and Alexa Fluor 800 Rabbit anti-Mouse (1:5000) (Molecular Probes, Life Technologies, Carlsbad, CA, USA) were incubated for 1 h at room temperature. Proteins were visualized using an Odyssey Infrared Imaging System (LI-COR Lincoln, Nebraska USA) according to the manufacturer’s instructions. Densitometry analyses were conducted using Quantity One software (Bio-Rad Laboratories, Mississauga, Canada). 

### 2.7. GO and Pathway Analysis

Gene Ontology Consortium (http://geneontology.org/) database (released 9 August 2018) was used to analyze the potential functions of mRNAs differentially expressed in untransfected and siRNA-transfected RANKL-induced RAW 264.7 cells. The PANTHER database (Protein Analysis Through Evolutionary Relationships, http://pantherdb.org) was used for the analysis as GO molecular function complete and GO biological process complete. Analysis was performed with the PANTHER Overrepresentation Test (released 5 December 2017), using a *Mus musculus* reference list (all genes in database). The test that was performed is the Fisher’s exact test with FDR correction. The default output was sorted by hierarchy of the categories. By default, only the categories with *p* value better than 0.05 were displayed. In the hierarchy view, the results were sorted by the fold enrichment of the most specific categories, with their parent terms (*p* value better than 0.05) indented directly below. Results of all *p* values have been displayed.

Protein network analysis was performed using Qiagen’s Ingenuity Pathway Analysis (IPA, Qiagen Redwood City, CA, USA) software.

### 2.8. Statistical Analysis

Data are expressed as mean ± S.D. of at least three independent experiments. Statistical significance between two groups was determined by a two-tailed Student’s *T* test. *p* < 0.05 was considered to indicate a statistically significant difference.

## 3. Results

### 3.1. Effects of NFATc1 Loss on Differentiation into Osteoclasts

To follow osteoclastogenesis in vitro, RAW 264.7 cells were stimulated with RANKL and observed for the formation of multinucleated cells. In the absence of RANKL stimulation, cells were mainly mono-nucleated and with a rounded morphology (Figure 1A, −/−), whereas, in the presence of RANKL stimulation, some multinucleated cells were observed among the cell population both in untransfected and in NC-siRNA transfected cells (Figure 1A, −/+ and NC/+). Instead, NFATc1-siRNA transfected cells showed only mono-nucleated cells (Figure 1A, NFATc1/+). To ensure that *NFATc1* had actually been silenced, the expression of both NFATc1-mRNA (Figure 1B) and protein (Figure 1C) were evaluated after one day of RANKL treatment by QPCR and western blot, respectively. 

### 3.2. Expression Profiles of Genes in Pre-Osteoclasts 

To dissect the pathway of NFATc1 and discover new molecules/transcription factors related to this pathway, we performed PCR array analysis. Total RNA extracted from untransfected pre-osteoclasts (−/− or −/+ RANKL) and transfected +RANKL (siRNA-NC or siRNA-NFATc1) was used to analyze the expression profiling of mouse transcription factors (TFs) and osteoporosis genes by PCR arrays. In detail, the first group of PCR array data came out of the analysis between untransfected cells +RANKL compared to untransfected cells -RANKL (named untransfected in the following); the second group of data came out of the analysis between transfected cells with siRNA-NC +RANKL compared to transfected cells with siRNA-NFATc1 +RANKL (named NFATc1-knockdown in the following). In total, the expression of 164 genes was analyzed and the heat-map profiles are shown (Figure 2A,B). The PCR array data from the two comparison groups were set according to a Venn diagram. The expression of 55 genes (Figure 2C) was significantly modified (≥2-fold) in untransfected cells, including 29 up-regulated (Figure 2D) and 26 down-regulated (Figure 2E) genes, respectively. Similarly, in NFATc1-knockdown cells the expression of 31 genes (Figure 2C) was significantly modified (≥2-fold), including 20 up-regulated (Figure 2D) and 11 down-regulated (Figure 2E) genes, respectively.

To identify NFATc1-target genes in RAW 264.7 macrophages, we conducted comprehensive expression profiling of untransfected and NFATc1-knockdown cells. The list of genes activated and repressed (≥2-fold) in untransfected cells are specified in Table 1, whereas the complete list of genes with the magnitude of changes are specified in Appendix A. In untransfected cells, the increase of *NFATc1* expression was confirmed based on the profiling data as well as quantitative PCR analysis (*p* < 0.05). The Gene Ontology (GO) analysis as biological process revealed the up-regulation of the expression of genes belonging to “development process” (*GATA3, Tfpa2a, Myf5, Pthr1, Tnfsf11*, *Gli1, Nog, Stat4, Ltbp2, Twist, Lta, GATA2*), “induction of apoptosis” (*Tnfsf11, Lta*), “cellular defense response” (*Tnfsf11, Stat4, Lta*), “mesoderm development” (*Ltbp2, Pthr1, Stat4, Twist1*) as well as “regulation of transcription RNA pol. II” (*Smad5, Hnf1a, Myc, NFATc1*). In contrast, genes belonging to “cell differentiation” (*Jun, JunD, Rb1, Ets2, Bmp2, Fgfr1*) as well as “nitrogen compound metabolic process” (*Mef2c, Jun, NFATc4, Rb1, JunD, Mef2b, Ets2, Egr1, Bmp2, Atf3*) and “response to endogenous stimuli” (*Timp2, Jun, JunD, Fgfr1*) were down-regulated, as expected (Figure 3A). In addition, GO analysis revealed significant enrichment of genes related to “development process” (*p =* 1.76 × 10^−7^), “cellular defense” (*p =* 6.48 × 10^−4^) and “regulation of transcription RNA pol. II” (*p =* 8.53 × 10^−6^); “response to endogenous stimuli” (*p =* 1.694 × 10^−4^), “cell differentiation” (*p =* 4.31 × 10^−5^), “metabolic process” (*p =* 1.95 × 10^−5^), “biosynthetic process” (*p =* 2.42 × 10^−6^) and “nitrogen compound metabolic process” (*p =* 3.92 × 10^−4^) (Figure 2A).

The list of genes activated and repressed (≥2-fold) in NFATc1-knockdown cells are specified in Table 2, whereas the complete list of genes with the magnitude of changes are specified in Appendix A. In NFATc1-knockdown cells, the inhibition of *NFATc1* expression was confirmed based on the profiling data as well as quantitative PCR analysis (*p* < 0.05). The analysis with GO as biological process revealed that the up-regulated genes mainly belonged to “cell differentiation” (*Myf5, Runx2, Myod1, Ets2* and *GATA2*), “system development” (*Myf5, Runx2, Myod1, Pthlh* and *GATA2*) and “development process” (*Gli1*, *Hand1, Alpl, Col1a2*, *Igf1* and *TIMP2*). In contrast, genes related to “osteoclast differentiation” and “bone resorption” (*NFATc1, CtsK, Acp5, Tnfsf11*) were down-regulated, as expected, as well as *Bmp2* and *Fgfr1* related to osteoblast differentiation (Figure 3B). In addition, GO analysis revealed significant enrichment of genes related to “development process” (*p* = 1.57 × 10^−6^), “cell differentiation” (*p* = 7.99 × 10^−5^), “cell-cell signaling” (*p* = 1.95 × 10^−4^), “osteoclast differentiation” (*p* = 1.49 × 10^−4^) and “bone resorption” (*p* = 1.04 × 10^−7^) (Figure 3B). 

We performed GO analysis as “molecular function” between the two groups (untransfected and NFATc1-knowdown), which indicated “binding” (GO:0005488) as the most representative category to include changes in the expression of genes (Figure 4). This category indicated molecules able to bind DNA, which is not surprising because most of the genes whose expression changed under the different conditions were indeed transcription factors. We noticed that there were no substantial differences in the percentage of genes belonging to the “binding” category between up- and down-regulated in both groups (Figure 4A,B). However, the amount of down-regulated genes in the “binding” category in NFATc1-knockdown cells was not significant (expected value 60%) (Figure 4B). Actually, the amount of down-regulated genes in NFATc1-knockdown cells grouped in different categories (GO molecular function analysis) indicated that there was no significant number of genes changing their expression (enrichment genes) in any categories showed. 

### 3.3. Validation of PCR Array Data and Gene Expression Analysis by QPCR and Western Blot

To validate our PCR array results, we selected some differentially expressed genes between the two groups (untransfected or NFATc1-knockdown) for QPCR or western blot analyses. As expected, untransfected cells showed an increase of NFATc1 mRNA and protein levels compared to the control (Figure 5A,H). Considering untransfected cells, the cytokine RANKL promoted a strong increase of *Myc* and *Acp5* gene expression in the PCR array (Table 1) and QPCR analyses (Figure 5A). On the contrary, *Jun*, *Esr1* and *Egr1* were down-regulated in both analyses (Table 1 and Figure 5B), whereas *Stat4, Smad5* and *Hnf1a* showed increased expression in PCR array (Table 1) while they did not change in QPCR analysis with respect to the control (Figure 5C). The *CtsK* gene expression did not show significant change in either analysis (Appendix A and Figure 5C). As expected, NFATc1-knockdown cells showed a reduction of NFATc1 mRNA and protein levels compared to the siRNA-NC-transfected cells (Figure 5E,H). Considering NFATc1-knockdown cells, the expression of transcription factor GATA2 was found strongly increased in PCR array (Table 2) as well as in QPCR analyses (Figure 5D), whereas the increased expression of *Runx2* (2.88-fold) found in the PCR array (Table 2) was not found in the QPCR analysis (Figure 5D). As expected, the expression of genes specific to osteoclast differentiation was greatly reduced (*Acp5, CtsK*) by an average of 70–75% in both analyses (Table 2 and Figure 5E). The expression of the transcription factors *Stat4*, *Hnf1a* and *Egr1* was down-regulated in QPCR analysis (Figure 5E) in accordance with PCR array data (Appendix A). The expression of *SMAD5, Esr1* and *Jun* did not significantly change in QPCR analysis (Figure 5F) compared to the control as well as in the PCR array (Appendix A). In addition, we analyzed other genes not included in the PCR arrays coding for molecules involved in cellular division, migration and fusion. Among these genes, *DC-STAMP, MMP9* and *OSCAR* showed a severe reduction of mRNA expression in NFATc1-knockdown cells and *DC-STAMP* and *MMP9* showed a strong induction in untransfected cells, while the expression of *RhoA* mRNA was slightly but significantly reduced in NFATc1-knockdown cells (by 30%) (Figure 5G). We also analyzed the expression of the transcription factors closely related to osteoclast differentiation processes and we again found opposite behaviour in the two conditions, as with both *MITF* and *FOS* (Figure 5G). By western blot, we analyzed NFATc2, DC-STAMP, MITF, RANK and TRAF6 and found that TRAF6 and RANK proteins were up-regulated in untransfected cells, whereas MITF and DC-STAMP proteins were down-regulated in cells NFATc1-knockdown cells (Figure 5H). 

### 3.4. Identification of Common and Unique Genes Targeted by NFATc1

To identify new potential NFATc1-target genes, we focused on 19 common genes that showed changes (up- or down-regulation) in the PCR arrays that were common between the two groups of cells (Table 3). Among them, 9 genes were down-regulated in untransfected and up-regulated in NFATc1-knockdown cells, 4 genes were up-regulated in untransfected and down-regulated in NFATc1-knockdown cells and finally 6 genes were regulated in the same way (up- or down-regulated) in both group of cells. We looked at the 9 genes that were down-regulated in untransfected cells and up-regulated in NFATc1-knockdown cells, since these genes were most likely to be new targets of NFATc1. These genes were: *Alox15, Alox5, Cnr2, Ets2, GATA2, Igf1, Lep*, *Spp1* and *Timp2*. Gene Ontology analysis (molecular function) revealed significant enrichment of genes related to “binding” (GO: 0005488), “catalytic activity” (GO.0003824) and “receptor activity” (GO: 0004872). Mechanistic network analysis by Ingenuity Pathway Analysis (IPA) software by Qiagen revealed a correlation among NFATc1, GATA2 and members of the STAT transcription factors family (STAT2 and STAT5) (Figure 6). 

### 3.5. GATA2 as a New Target of NFATc1

GATA2 transcription factor is considered an inducer to reprogram cell pluripotency [15]. Our PCR array data showed that the gene encoding for *GATA2* was significantly up-regulated (32.4-fold) in NFATc1-knockdown cells. Accordingly, *GATA2* mRNA (Figure 5D) and protein levels (Figure 7A) in NFATc1-knockdown cells were significantly up-regulated (*p* < 0.001). After 72 h, due to the transient transfection, NFATc1 protein levels were rescued in NFATc1-knockdown cells, while GATA2 protein levels were reduced with respect to the siRNA-NC control cells (Figure 7A). In order to clarify the possible role of GATA2, we analyzed the expression of STAT6, a transcription factor known to interact with GATA2 [16] and involved in macrophage differentiation. As a result, in Figure 7B we saw a significant increase in STAT6 protein levels and, to confirm that it actively transcribed, we analyzed one of its targets as an sample *Arginase1* (Arg1) in NFATc1-knockdown cells (Figure 7C). STAT6 has an important role in the transcription of some cytokines involved in macrophage differentiation [17]. Therefore, we analyzed the expression of *IL-10*, *IL-13* and *TNF-α* mRNAs by QPCR in NFATc1-knockdown cells. As shown in Figure 7D, *IL-13* mRNAs were significantly down-regulated, whereas *IL-10* and *TNF-α* expression did not change in NFATc1-knockdown cells Figure 7D). Altogether, these results suggest that GATA2 expression depends on the presence/absence of NFATc1 in macrophages and could be involved in their choice of differentiation. 

## 4. Discussion

Osteoclasts are derived from hematopoietic stem cell (HSCs) and later from monocyte-macrophage lineage through the action of two cytokines, the M-CSF and RANKL [5]. The RAW 264.7 murine macrophage cell line consists of a population of pre-osteoclasts, which need only RANKL-stimulation to differentiate in osteoclasts in vitro over a four day period. As already reported, after RANKL-induction the first step is the expression of NFATc1 transcription factor. As a master regulator, NFATc1 may indeed regulate expression of many other genes [10]. However, this transcription factor is expressed in many different cell types where it does not necessarily imply differentiation into osteoclasts. The comprehension of the mechanisms that regulate initial processes of osteoclastogenesis is of paramount importance to understanding the process of osteoclast differentiation, and in turn for managing many bone-related diseases, such as osteoporosis, osteopetrosis and rheumatoid arthritis (AR) [9,18,19]. 

In this study, we utilized PCR array analysis to identify downstream targets of NFATc1 to clarify steps involved in pre-osteoclasts differentiation. We compared the transcriptomes of RANKL-induced pre-osteoclasts with defective or complete expression of NFATc1. We found that in untransfected cells, the cytokine modulated expression of many genes grouping in different functional categories. Gene Ontology analysis (as biological processes) grouped the up-regulated genes into “development process” (12 genes *p* 1.76 × 10^−7^) and “regulation of transcription RNA pol. II” (7 genes *p* 8.53 × 10^−6^). On the contrary, in NFATc1-knockdown cells, up-regulated genes were mainly grouped into the categories “development process” (9 genes *p* 1.57 × 10^−6^) and “cell differentiation” (5 genes *p* 7.99 × 10^−5^). Instead, the genes down-regulated in untransfected cells were grouped into “metabolic process” (18 genes *p* 1.95 × 10^−5^), “biosynthetic process” (11 genes *p* 2.42 × 10^−6^), “nitrogen compound metabolic process” (10 genes *p* 3.92 × 10^−4^) and “cell differentiation” (6 genes *p* 4.31 × 10^−5^). Most of these genes are involved in the “housekeeping” activities of many cell types. On the contrary, down-regulated genes in NFATc1-knockdown cells belong mainly to groups related to “cell signaling” (4 genes *p* 1.95 × 10^−4^), “bone resorption” (3 genes *p* 1.04 × 10^−7^) and “osteoclast differentiation” (2 genes *p* 1.49 × 10^−4^), as expected in cases of impairment of the key regulator for osteoclast transcriptional program. We focused on the “developmental processes” and “cell differentiation” related genes, since the identification of these genes could help to elucidate the molecular mechanism that underlies osteoclast differentiation. 

Venn analysis revealed that 19 genes showed changes in their expression levels in both groups (untransfected and NFATc1-knockdown). In detail, we found genes that were up-regulated in one group and down-regulated in the other, and genes that were up-regulated or down-regulated in both groups. Among them, there were genes that changed expression level dramatically and genes that showed very mild changes. As an example, *Acp5* was up-regulated in untransfected cells and down-regulated in NFATc1-knockdown cells (fold changes: 41.89 and −3.06, respectively). This result was expected, since *Acp5* codes for TRAP, which is a hallmark of osteoclast differentiation. Another interesting gene that changed in the two conditions was the transcription factor *Jun*. This gene was significantly down-regulated (fold change: –11.86) in untransfected cells, while it was just under the exclusion threshold in NFATc1-knockdown cells (1.66). Our GO analysis grouped this gene in the “cell differentiation” category together with *Ets2* (−2.53 and 2.76), *Runx2* (1.05 and 2.88) and *Myod1* (1.79 and 3.21), showing the same trend in the two groups. On the contrary, bone morphogenetic protein 2 (Bmp2) was down-regulated in the two groups (fold changes: −4.15 and −3.69), which was also expected. In fact, Bmp2 is a hallmark of osteoblast differentiation and it is likely that NFATc1 does not regulate it. 

Among the up-regulated genes in untransfected cells, *Myc* appeared interesting since it strongly increased in our PCR array (108-fold) and was validated by QPCR. *Myc* is a proto-oncogene known for controlling proliferation, differentiation, apoptosis and cancer. However, it has been reported that *Myc* is correlated with osteoclasts differentiation induced by RANKL, and its dominant negative isoform reduces the expression of osteoclast marker proteins such as CtsK and TRAP [20]. In our bioinformatics analysis, *Myc* belonged to the GO group “regulation of transcription RNA pol. II” together with NFATc1. This group collected transcription factors that could participate in osteoclasts differentiation. Accordingly, our data showed a strong increase of *Myc* expression in the initial phase of differentiation in untransfected cells but not in NFATc1-knockdown cells, correlating *Myc* expression to NFATc1 levels. Recently, it has been demonstrated that *Myc* has a role in the induction of estrogen receptor–related receptor **α** (ERR**α**), a nuclear receptor that cooperates with NFATc1 to drive osteoclastogenesis [21]. 

Another transcription factor up-regulated in untransfected cells and down-regulated in NFATc1-knockdown cells was *MITF*, which we analyzed regardless of the PCR arrays data. MITF is a family of transcription factors expressed in a tissue- and cell lineage-specific manner; among them MITF-E is expressed in monocyte-macrophage cell lineage and osteoclasts. It has been reported that miR-155 inhibits osteoclast differentiation by targeting MITF [22]. Data obtained by QPCR and western blot showed that MITF-mRNA and protein decreased in NFATc1-knockdown cells (Figure 5G–H). Lu et al. [23] found that MITF silencing did not influence NFATc1 expression or activation, and that MITF up-regulation did not have positive influence on NFATc1 expression level. These results suggest that MITF functions downstream of NFATc1 within RANKL signaling, even if it was found that MITF up-regulation was sufficient to induce osteoclast markers such as TRAP, DC-STAMP, CtsK and ATP6V0 [23]. Accordingly, our data point to MITF as a target of NFATc1, since NFATc1-depletion inhibits MITF expression. Ultimately, the role of MITF is to promote the amplification of the NFATc1 signaling during osteoclastogenesis [23], whereas NFATc1 is critical for MITF expression (Figure 5F,G). Therefore, it might be proposed that MITF is an inducer for the osteoclast differentiation acting in concert with NFATc1. In fact, as we have already discussed, NFATc1 is ubiquitous and is involved in many differentiation programs other than osteoclastogenesis, whereas MITF-E is present only in the osteoclasts. 

Another interesting gene is *GATA2*, which is up-regulated in the two groups of cells but with strong differences. The level of its expression is just above the exclusion threshold (2.59-fold) in untransfected cells, but is strongly up-regulated in NFATc1-knockdown cells (32.40-fold). We validated these results finding increased expression levels of both mRNA and protein in NFATc1-knockdown cells (Figure 5D and Figure 7A). Interestingly, the GATA2-mRNA levels were inversely correlated with those of NFATc1, suggesting the existence of a cross-regulation between NFATc1 and GATA2 in RAW 264.7 cells. In our bioinformatics analyses, GATA2 belonged to the GO group of “cell differentiation” and, most importantly, it was a central node in the protein network analysis by IPA (Figure 6), closely related to NFATc1 and STAT family members. GATA2 is considered an inducer for the reprogramming of cell pluripotency and belongs to a family of six transcription factors with well-established roles in cellular differentiation. GATA1, GATA2 and GATA3 are expressed in hematopoietic cells [24,25,26], whereas GATA4, GATA5 and GATA6 regulate cell differentiation in non-hematopoietic cells [27,28,29]. GATA1 and GATA2 are often expressed in a mutually antagonistic manner, regulating the differentiation of hematopoietic stem cells (HSCs) at different times [26,30]. In fact, GATA2 acts very early in the differentiation of bone marrow–derived macrophage (BMM)–HSCs and is down-regulated during lineage commitment, whereas GATA1 interferes in the last steps of the differentiation [30]. In addition, GATA2 is required to generate osteoclast progenitors [31], while GATA1 is dispensable for osteoclastogenesis [32]. Therefore, the GATA2–GATA1 ratio in hematopoietic progenitors controls lineage divergence between osteoclasts and erythrocytes/megakaryocytes. In agreement, our PCR array data showed the GATA2–GATA1 ratio was 2.59:1 in untransfected cells and 32.40:1 in NFATc1-knockdown cells, as expected in the case of lineage divergence towards osteoclasts. In addition, it should be taken into account that GATA2, but not GATA1, is critical in maintaining the differentiated state of mast cell/basophil as well as eosinophil lineages [33]. In fact, deletion of GATA2 in bone marrow mononuclear cells (BMMCs) results in an impairment of mast cell–like phenotype with a strong expression of macrophage markers, such as CD11b- and/or Ly6G/C [34]. GATA2 is also crucial for maintaining the expression of the hallmarks of mast cell/basophil phenotypes, for example, *Alox5,* which is commonly expressed in both cell types [35]. In agreement, our array data showed that *Alox 5* and *Alox 15* increased mRNA expression levels (5.25 and 3.83, respectively) in NFATc1-knockdown cells parallel to *GATA2* up-regulation, while, on the contrary, they were strongly down-regulated (−36.52 and −47.76, respectively) in untransfected cells. Our data strongly suggests that GATA2 is a downstream target of NFATc1, and we might speculate that NFATc1 possibly represses its expression through miRNA induction.

Many cytokines are crucial for the differentiation of pre-osteoclasts, such as IL-1, IL-6, IL-15 and TNF-α, whereas others suppress osteoclastogenesis including IL-3, IL-4, IL-10 and IFN [36]. Accordingly, our PCR arrays data showed that the pro-osteoclastogenesis cytokine *IL-15* increased (2.50-fold) in untransfected cells because of RANKL-stimulation (Table 1). Cytokines are mainly produced by T lymphocytes, although IL-6, IL-10, IL-13 and TNF-α are also produced by mast cells and monocyte-macrophages in which they induce the differentiation [37]. In addition, the pro-inflammatory cytokines IL-6 and TNF-α are considered hallmarks of activated macrophage phenotype M1, whereas IL-10 and IL-13 are hallmarks of alternatively activated macrophage phenotype M2 (osteoclast-like) [38]. Our PCR array data showed that *IL6* expression level is under the exclusion threshold (≥2) in both treatments (Appendix A). Further, we analyzed the mRNA expression levels of *IL-10, IL-13* and *TNF-α* cytokines. Our results indicated that *TNF-α* and *IL-10* expression levels did not significantly change in NFATc1-knockdown cells with respect to the untransfected cells, whereas *IL-13* mRNA was significantly inhibited in NFATc1-knockdown cells (Figure 7D). The cytokine IL-4 has anti-osteoclastogenesis activity through binding with its receptor (IL4-R) and activation of the transcription factor STAT6 [39]. In agreement, our results indicate that STAT6 protein levels increased in NFATc1-knockdown cells, suggesting anti-osteoclastogenesis activity. Altogether, these data suggest that NFATc1 ablation caused GATA2 and STAT6 activation, but did not inhibit the expression of pro-osteoclastogenesis cytokines (TNF-α and IL-6), whereas it caused an impairment of the expression of some cytokines (IL-13) associated with macrophage differentiation of the M2 phenotype (osteoclast-like). 

Ultimately, the balances and order of expression of multiple transcription factors, rather than the expression level of a single factor, such as *GATA2*, *STAT*6 and *NFATc1*, are important for the specification of cellular fate. This is not surprising since it has been reported that exactly these factors—GATA2, STAT6 and NFATc1—are tightly controlled for the genome-wide occupancy and consequent transcriptional programs that could be activated, such as in the case of endothelial cell activation [40]. 

Nevertheless, our data suggest that GATA2 could be one of the important factors that determine differentiation towards macrophages in the absence of NFATc1, even if the cellular fate in the presence of GATA2 up-regulation is presently unknown. However, the observed activation of the transcription factors STAT6 and GATA2 in the absence of NFATc1 are worthy of further studies.

Unraveling the molecular program that drives the differentiation and function of osteoclasts is important for a better understanding of many diseases linked to defective osteoclastogenesis. 

## Figures and Tables

**Figure 1 cells-08-00131-f001:**
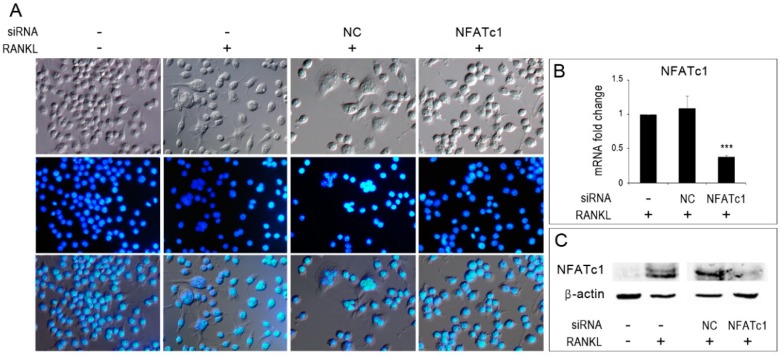
Inhibition of osteoclastogenesis by silencing of NFATc1. Untransfected, siRNA-non correlated (NC) and siRNA-NFATc1 transfected cells were cultured with RANKL (50 ng/mL) for 24 h. Control untransfected cells were cultured without RANKL. (**A**) Cells were fixed, stained with DAPI (which stains the nuclei blue) and observed by DIC (upper row) and immunofluorescence (middle row) microscopy. Bottom row shows merged images. (**B**) Quantitative PCR (QPCR) of *NFATc1*. mRNA expression was presented as relative values to those expressed in untransfected cells (−/+RANKL) arbitrarily set at 1.0. The results shown are the means ± SD of two experiments (each of which was performed in triplicate). *** *p* < 0.005. (**C**) Western blot of NFATc1 protein in untransfected (−/− RANKL) and (−/+ RANKL), siRNA-NC and siRNA-NFATc1 transfected cells (+RANKL). The data shown represent two independent experiments with comparable outcomes.

**Figure 2 cells-08-00131-f002:**
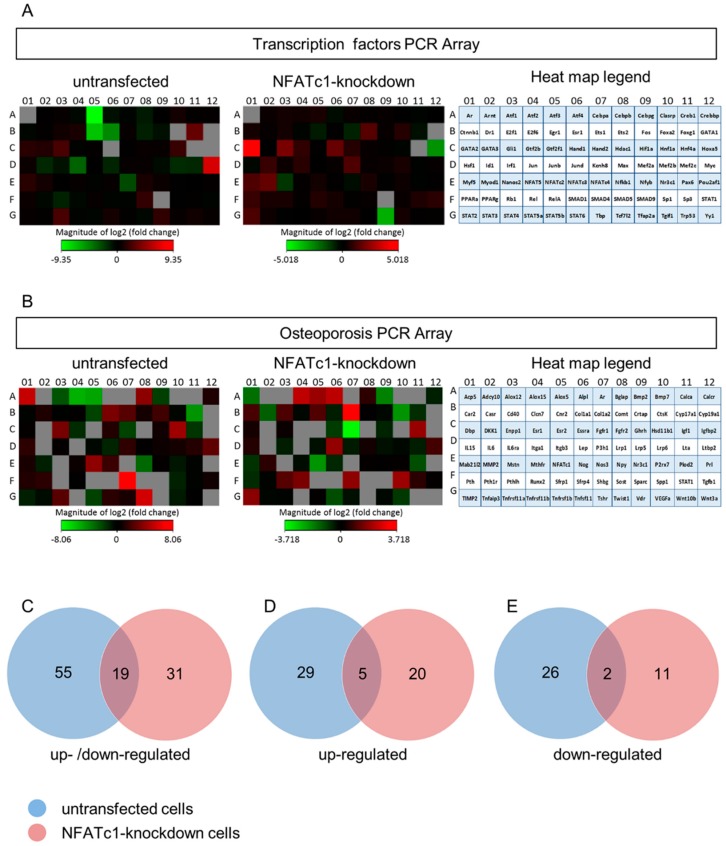
Expression profiles of transcription factors and osteoporosis PCR arrays in untransfected and NFATc1-knockdown RAW 264.7 cells. PCR arrays for transcription factors and osteoporosis were performed. The cluster heat maps for the expression of all genes from (**A**) transcription factors and (**B**) osteoporosis PCR arrays were shown for untransfected cells (left) and NFATc1-knockdown cells (right). The red squares represent up-regulated genes, the green ones down-regulated genes, the black ones unchanged genes and the grey ones are technically unacceptable data. These latter were not considered in the analysis of our results. Differentially expressed genes between untransfected and NFATc1-knockdown cells, RANKL-induced. (**C**) Up- and down-regulated genes, (**D**) up-regulated genes, (**E**) down-regulated genes.

**Figure 3 cells-08-00131-f003:**
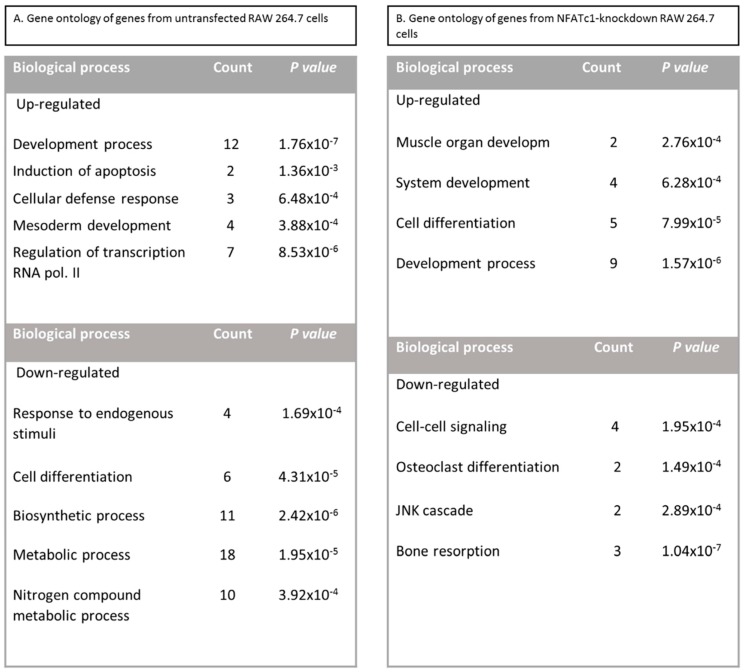
Gene Ontology analysis as biological function of untransfected and NFATc1-knockdown RAW 264.7 cells. (**A**) Untransfected cells, RANKL-induced. (**B**) NFATc1-knockdown cells, RANKL-induced. The count represents the number of genes showing variation in their expression levels involved in the annotated biological process. The *p* value denotes the significance of Gene Ontology (GO) term enrichment in the differentially expressed mRNAs based on the PANTHER classification system. This analysis allowed us to determine the biological pathways for which a significant enrichment of differentially expressed genes existed (*p* < 0.05) calculated with Fischer’s exact test with FDR multiple test correction.

**Figure 4 cells-08-00131-f004:**
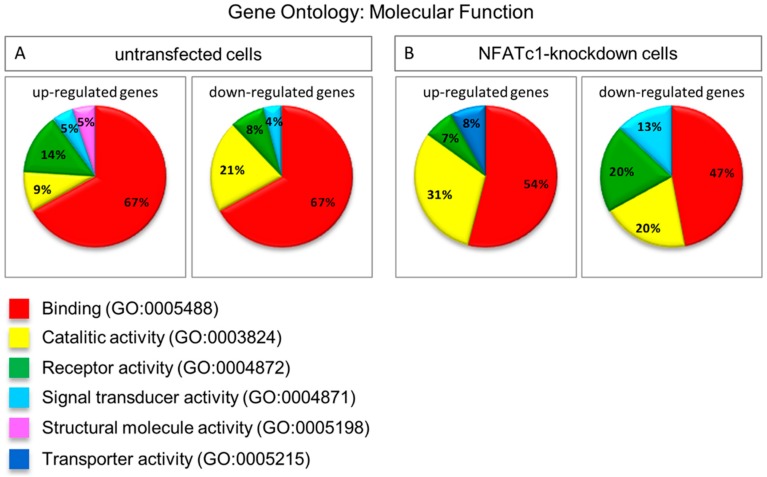
Gene Ontology analysis as molecular function of untransfected and NFATc1-knockdown cells. Gene Ontology analysis as molecular function represented by pie chart (**A**) untransfected cells, RANKL-induced; and (**B**) NFATc1-knockdown cells, RANKL-induced.

**Figure 5 cells-08-00131-f005:**
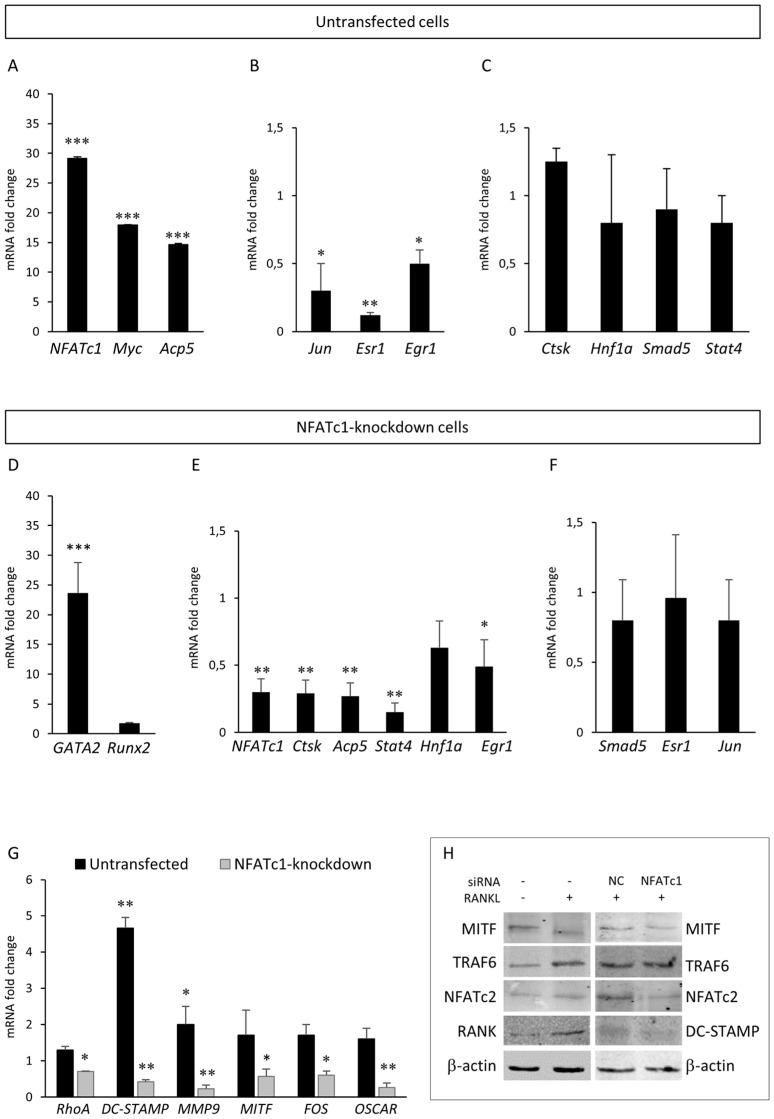
Effects of NFATc1 induction or depletion on different genes involved in the RANK–RANKL pathway. Cells were untransfected and RANKL-induced. QPCR results of (**A***) NFATc1, Myc* and *Acp5*; (**B**) *Jun, Esr1* and *Egr*1; and (**C**) *CtsK, Hnf1a, Smad5* and *Stat4*. Cells were NFATc1-knockdown and RANKL-induced. QPCR results of (**D**) *GATA2* and *Runx2*; (**E**) *NFATc1, CtsK, Acp5, Stat4, Hnf1a* and *Egr1*; (**F**) *Smad5, Esr1* and *Jun*; and (**G**) *RhoA, DC-STAMP, MMP9, MITF, FOS* and *OSCAR*. The mRNA expression levels were presented as relative values to those expressed in siRNA-NC transfected or untransfected cells arbitrarily set at 1.0. The results shown are the means ± SD of three experiments (each of which was performed in triplicate). * *p* < 0.05, ** *p* < 0.01 and *** *p* < 0.001 versus the control. (**H**) Western blot results of MITF, TRAF6, NFATc2, DC-STAMP and RANK proteins in untransfected and NFATc1-knockdown cells, RANKL-induced. The data shown represents two independent experiments with comparable outcomes.

**Figure 6 cells-08-00131-f006:**
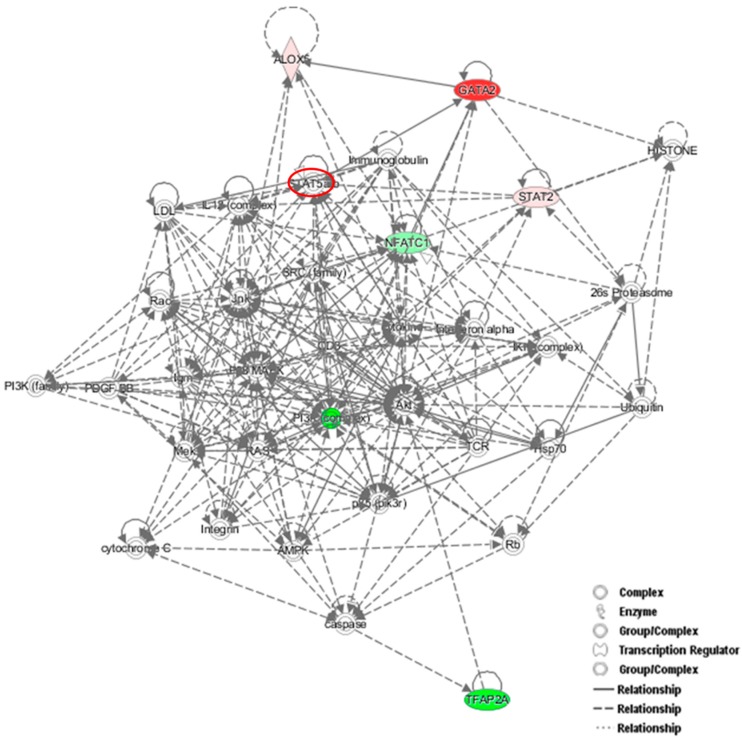
Protein network analysis using Ingenuity Pathway Analysis (IPA). Proteins identified in the dataset are highlighted in red or in green when they exhibit a higher or a lower content in NFATc1-knockdown cells compared to untransfected cells. The intensity of the node color indicates the expression level or degree of regulation. Genes in uncolored notes were not differentially expressed in our experiment and were integrated into the computationally generated networks based on the evidence stored in the IPA knowledge memory, which indicated relevance to this network.

**Figure 7 cells-08-00131-f007:**
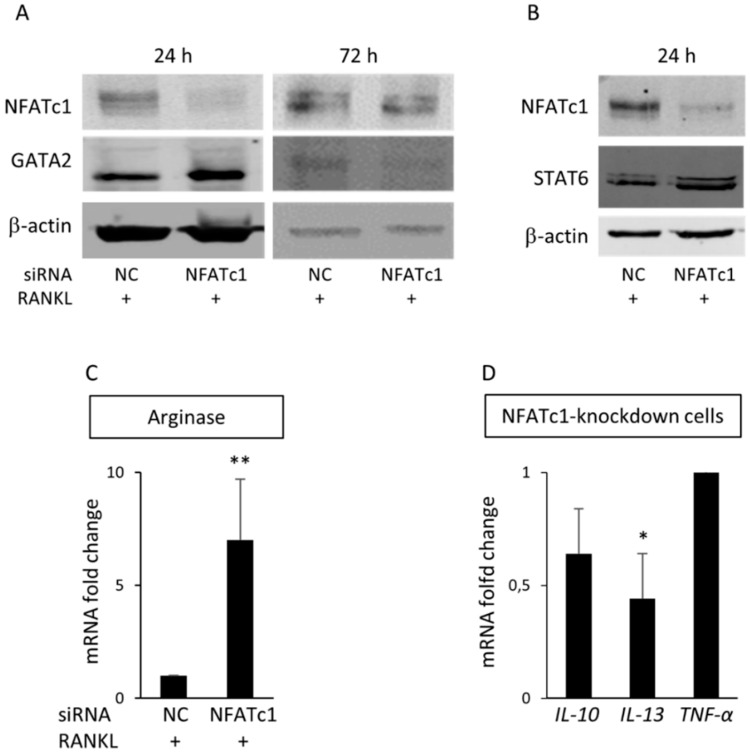
GATA2 is a new target of NFATc1. Cells were transfected with siRNA-NC and siRNA-NFATc1 for 24 and 72 h before protein expression levels were analyzed. (**A**) Western blot of NFATc1 and GATA2 after 24 and 72 h. (**B**) Western blot of NFATc1 and STAT6 after 24 h. β-actin was used as loading control. The data shown represent two independent experiments with comparable outcomes. (**C**) QPCR of *Arginase 1*. (**D**) QPCR of *IL-10, IL-13* and *TNF-α*. mRNA expression is presented as relative values to those expressed in siRNA-NC transfected cells arbitrarily set at 1.0. The results shown are the means ± SD of three experiments (each of which was performed in triplicate). * *p* < 0.05 ** *p* < 0.01.

**Table 1 cells-08-00131-t001:** List of genes from untransfected RAW 264.7 cells (genes ≥ 2-fold).

Gene Symbol	Fold Change	Gene Symbol	Fold Change
Acp5	41.89	Alox12	−6.37
Bglap	13.96	Alox15	−47.76
Col1a1	9.77	Alox5	−36.52
Col1a2	5.00	Atf3	−652.73
Crtap	3.49	Bmp2	−4.15
Enpp1	9.27	Cebpb	−2.14
Esrra	2.12	Cnr2	−4.46
GATA2	2.59	Cy17a1	−26.60
GATA3	2.23	Egr1	−144.98
Gli1	4.79	Esr1	−32.59
Hsd11b1	19.75	Ets2	−2.53
Hnf1a	3.59	Fgfr1	−4.50
Foxg1	8.82	Id1	−2.89
IL15	2.50	Igf1	−3.43
Myc	108.70	Jun	−11.85
P3h1	2.61	JunD	−2.72
Myf5	2.58	Lep	−2.42
Lta	2.22	Mef2b	−3.15
Ltbp2	6.05	Mef2c	−2.98
NFATc1	14.63	MMP2	−2.23
Nog	3.07	NFATc4	−6.64
Pthr1	2.38	Npy	−9.74
Shbg	266.84	Rb1	−2.65
Smad5	3.14	Spp1	−2.27
STAT4	5.18	Timp2	−2.22
Tfap2a	3.09	Tnfrsf1b	−4.27
Tnfrsf11a	5.26		
Tnfsf11	2.42		
Twist	51.33		
	29		26

**Table 2 cells-08-00131-t002:** List of genes from NFATc1-knockdown RAW 264.7 cells (genes ≥ 2-fold).

Gene Symbol	Fold Change	Gene Symbol	Fold Change
Alox15	5.25	Acp5	–3.06
Alox5	3.83	Bmp2	–3.69
Alpl	5.18	Casr	–2.61
Cnr2	2.04	CtsK	–4.51
Col1a2	13.16	Fgfr1	–12.61
Ets2	2.76	Hoxa5	–7.29
GATA2	32.4	Nanos2	–2.12
Gli1	4.1	NFATc1	–4.46
Igf1	3.72	Plood2	–2.26
Hand1	3.33	Tfap2	–10.74
Kcnh8	2.52	Tnfsf11	–2.22
Lep	3.51		
Myf5	2.54		
Myod1	3.21		
Pthlh	2.10		
Runx2	2.88		
Shbg	3.40		
Spp1	3.86		
STAT2	2.44		
Timp2	3.26		
	20		11

**Table 3 cells-08-00131-t003:** List of common genes that changed (≥2-fold).

Gene Symbol	Untransfected	NFATC1-Knockdown
Acp5	41.89	−3.06
Alox15	−47.76	5.25
Alox5	−36.52	3.83
Bmp2	−4.15	−3.69
Cnr2	−4.46	2.04
Col1a2	5	13.16
Ets2	−2.53	2.76
Fgfr1	−4.5	−12.61
GATA2	2.59	32.4
Gli1	4.79	4.1
Igf1	−3.43	3.72
Lep	−2.42	3.51
Myf5	2.58	2.54
NFATc1	14.63	−4.46
Shbg	266.84	3.4
Spp1	−2.27	3.86
Tfap2a	3.09	−10.74
Timp2	−2.22	3.26
Tnfsf11	2.42	−2.22
19

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
