# Peer review of "Gene Expression Profiling of NFATc1-Knockdown in RAW 264.7 Cells: An Alternative Pathway for Macrophage Differentiation"

_cells, 2019, doi:10.3390/cells8020131_

Reviewer 1 Report

NFATc1 is a master transcription factor in osteoclast formation and activation. Russo et al., examined expression profiles of untreated and NFATc1 knockdown RAW264.7 cells, pre-osteoclasts, after RANKL induction to clear the molecular pathway underlying osteoclast differentiation. Their result by GO and IPA mechanistic analysis identified GATA2, transcription factor involved in NFATc1 pathway. However, there are many mistakes through the whole manuscript. The authors should write their manuscript more carefully.

The author should at least revise the following points in the manuscript.

Major points

1. The authors found that GATA2 was up-regulated in NFATc1 knock-down cells. It suggested that GATA2 is involved in NFATc1 signaling pathway. Furthermore, NFATc1 knockdown increased STAT6 protein expression level in RAW264.7 cells. The purpose of this study is to find out molecular pathway in pre-osteoclast formation. Then, the authors should reveal the effect of GATA2 or STAT6 for osteoclast formation by knockdown or overexpression assay.

2. In Fig4H, the authors showed MITF, TRAF6, NFATc2 and DC-STAMP protein expression in untransfected cells and NC siRNA transfected cells treated with RANKL. However, the expression patterns of these proteins were not equal in both cells in western blot assay. I think the expression of these proteins in untransfected cells treated with RANKL should be similar to NC siRNA transfected cells treated with RANKL The authors should explain the reason of difference of expression patterns.

3. In Fig 6D, The authors showed IL-13 mRNA is downregulated in siRNA-NFATc1 transfected cells compared to in NC siRNA transfected cells. I think this change should be measured by ELISA.

Minor points

1. In Fig 1 A and B, the authors should describe what left panel and right panel represented.

2. In Fig 2, Panel (A) and Panel (B) are overlapped. The authors should revise it.

3. In Table2, the column of Fold change are disordered. The authors should revise it.

4. What present for C) and D) in legend of Fig. 3?  The authors should revise them.

5. In references, there are many mistakes about reference style, author’ name, etc.

The authors should revise them.

Author Response

Reviewer 1

Major points

1. The authors found that GATA2 was up-regulated in NFATc1 knock-down cells. It suggested that GATA2 is involved in NFATc1 signaling pathway. Furthermore, NFATc1 knockdown increased STAT6 protein expression level in RAW264.7 cells. The purpose of this study is to find out molecular pathway in pre-osteoclast formation. Then, the authors should reveal the effect of GATA2 or STAT6 for osteoclast formation by knockdown or overexpression assay.

 1. Concerning the reviewer request, the effects of GATA2 knockdown on osteoclast formation has already been described in literature. Indeed, it has been shown that the loss of GATA2 caused defects in the generation of osteoclast progenitors, but not in the formation of mature osteoclasts (Yamane T. et al. 2000; Wei W. et al. 2011), thus suggesting that GATA2 is required for an early stage of osteoclastogenesis, but not for the terminal differentiation stage. In addition, late GATA2 ablation allowed in any case the generation of a small number of osteoclasts (Wei W. et al. 2011).

Furthermore, the expression of GATA2 in hematopoietic staminal cells (HSCs) is required for the expansion of multipotent hematopoietic cells and the formation of mast cells, but it was also found to be dispensable for the terminal differentiation of erythroid cells and macrophages (Tsai FY, Orkin SH. 1997).

Our data indicate that GATA2 overexpression is found in NFATc1-knockdown cells, which show inhibition of both the expression of specific osteoclast hallmarks (CtsK, MITF, Oscar, DC-STAMP, Acp5, MMP9) and the formation of multi-nucleated osteoclasts.

Actually, as well understood by the reviewer, the purpose of this manuscript was investigating the role of NFATc1 for the generation of osteoclasts from macrophages and discover its new targets, if any, by knocking down it. Really, we found some direct/indirect targets and especially we found that the NFATc1 knockdown could direct macrophages towards a different way of differentiation, in which GATA2 seems to be involved.  

Of course, we think that it would be interesting to study the overexpression/knockdown of GATA2 for future investigations, to better understand its role in macrophages differentiation, but we do think it would be off-topic to perform these kind of experiments here considering the purpose of this manuscript as previously quoted.

2. In Fig4H, the authors showed MITF, TRAF6, NFATc2 and DC-STAMP protein expression in untransfected cells and NC siRNA transfected cells treated with RANKL. However, the expression patterns of these proteins were not equal in both cells in western blot assay. I think the expression of these proteins in untransfected cells treated with RANKL should be similar to NC siRNA transfected cells treated with RANKL The authors should explain the reason of difference of expression patterns.

 2. The reviewer is right, as the protein levels of MITF, TRAF6, NFATc2 are expected to be similar in untrasfected cells (+RANKL) and NC-siRNA transfected cells (+RANKL). Actually, the differences observed by the reviewer arise from the analysis of different Western blots for the two group of proteins that thus should not be compared. However, for the significance of the results, what is important is the presence of the reference protein in each group of Western blot, i.e. β-actin. Nevertheless, we have lightened the photos in figure 4H that perhaps were too dark, and now the protein levels of MITF, TRAF6, NFATc2 appear almost similar. Furthermore, we have to underline that the protein levels of RANK were analysed only in untransfected cells, while those for DC-STAMP were analysed only in transfected cells.

3. In Fig 6D, The authors showed IL-13 mRNA is downregulated in siRNA-NFATc1 transfected cells compared to in NC siRNA transfected cells. I think this change should be measured by ELISA.

 3.  We think that the sensitive technique of qPCR revealing the mRNA expression is sufficient here. Indeed, measuring the levels of IL-13 protein by ELISA would not add any interesting information for the purpose of this manuscript.

Minor points

1. In Fig 1 A and B, the authors should describe what left panel and right panel represented.

1. We added a new Fig.1 therefore Fig.1 became Fig.2. As suggested by the reviewer in Fig2 A and B now we describe what left panel and right panel represented.

2. In Fig 2, Panel (A) and Panel (B) are overlapped. The authors should revise it.

2. As suggested by the reviewer we revised Fig.2 that now became Fig3.

3. In Table2, the column of Fold change are disordered. The authors should revise it.

3. As suggested by the reviewer we revised Table 2

4. What present for C) and D) in legend of Fig. 3?  The authors should revise them.

 4. As suggested by the reviewer we revised legend of Fig.3 that now became Fig4.

5. In references, there are many mistakes about reference style, author’ name, etc.

The authors should revise them.

5. As suggested by the reviewer we revised references

Reviewer 2 Report

In this manuscript, the authors use rt-PCR to identify transcription factors whose expression changes with macrophage differentiation into osteoclasts, in the presence or absence of NFATc1. However, their analysis is largely descriptive and lacks any functional testing, making the manuscript in its current form of limited value to the field.

Major concerns:

The authors do not show any data demonstrating that the treatment with RANKL or the knockdown of NFATc1 affected differentiation into osteoclasts. Histology or western blot analysis demonstrating that their conditions affected differentiation would strengthen their argument that the changes in gene expression observed are biologically relevant.

The authors repeatedly state that they used a microarray approach to examine changes in gene expression. Describing the RT2 profiler PCR array as a microarray is misleading. In addition, they do not indicate whether they used different primer sequences to validate differentially expressed genes.

There is no key to the heat maps presented in Figure 1, making them of limited value to the reader.

The authors should consider comparing the gene expression in untransfected, RANKL treated cells, with the cells transfected with si-NC and treated with RANKL in order to demonstrate that the changes observed between untransfected cells +/- RNAKL, and in RANKL treated si-NC and si-NFATc1 cells are real and not artifacts.

The western blots in Figure 4 are very dark and not very convincing.

The authors propose that genes down regulated with RANKL treatment in untransfected cells, but up regulated in NFATc1 knockdown cells are direct targets of NFATc1, but they do not directly test this, nor do they test whether these genes are required for osteoclast differentiation. They further argue that GATA2 and STAT family members are at the center of this pathway based on an IPA network. However, as this analysis is based on a very small set of differentially expressed genes, it is unclear how biologically relevant this conclusion is.

The manuscript has several grammar and formatting issues, including non-standard English (such as up- and down- expressed, instead of up regulated and down regulated), changes in line spacing, and seemingly random italicization.

Author Response

Reviewer 2

Major concerns:

1. The authors do not show any data demonstrating that the treatment with RANKL or the knockdown of NFATc1 affected differentiation into osteoclasts. Histology or western blot analysis demonstrating that their conditions affected differentiation would strengthen their argument that the changes in gene expression observed are biologically relevant.

1. As suggested by the reviewer now we have added a new Fig.1 with immunofluorescences showing peculiar cells fusion in the presence of RANKL in untransfected and NC-siRNA transfected cells and the inhibition of cells fusion in cells transfected with NFATc1-siRNA. Moreover, qPCR and Western blot analyses are shown in Fig.1B and C to demonstrate the down-regulation of both the NFATc1 mRNA and the protein expression in NFATc1-siRNA transfected cells.

2. The authors repeatedly state that they used a microarray approach to examine changes in gene expression. Describing the RT2 profiler PCR array as a microarray is misleading. In addition, they do not indicate whether they used different primer sequences to validate differentially expressed genes.

2. As suggested by the reviewer the word microarray was substituted by the more correct “PCR array” throughout the text.

As stated in Materials and methods we performed PCR arrays from the Qiagen company; also primers used to perform qPCR gene expression validation were bought by the same company. We checked the differences among primers used to validate differentially expressed genes and we found that out of 12 validations, 7 primers were different. In particular, primers for NFATc1, Myc, Acp5, Esr1, CtsK, Runx2 and GATA2 are different.

3. There is no key to the heat maps presented in Figure 1, making them of limited value to the reader.

3. As suggested by the reviewer now we have added a key to the heat maps presented in old Figure 1 (new Fig2).

4. The authors should consider comparing the gene expression in untransfected, RANKL treated cells, with the cells transfected with si-NC and treated with RANKL in order to demonstrate that the changes observed between untransfected cells +/- RNAKL, and in RANKL treated si-NC and si-NFATc1 cells are real and not artifacts.

4. As suggested by the reviewer now we have compared the gene expression of NFATc1 induced by RANKL between untransfected and transfected cells with NC-siRNA. The results of q-PCR analysis are showed in the new Fig.1B together with NFATc1 Western blots of the same treatments (old Fig.4H).

5. The western blots in Figure 4 are very dark and not very convincing.

5. As suggested by the reviewer we have lightened the photos in Fig.5H (old Fig.4H) improving the resolution of Western blot results.

6. The authors propose that genes down regulated with RANKL treatment in untransfected cells, but up regulated in NFATc1 knockdown cells are direct targets of NFATc1, but they do not directly test this, nor do they test whether these genes are required for osteoclast differentiation. They further argue that GATA2 and STAT family members are at the center of this pathway based on an IPA network. However, as this analysis is based on a very small set of differentially expressed genes, it is unclear how biologically relevant this conclusion is.

6. We agree with the reviewer as to whether, with the data shown here, we cannot assert that any of the up/down expressed genes are “direct” targets of NFATc1 gene. Indeed, we only suggest that GATA2/STAT genes are possible “new” targets of NFATc1, but we do not sustain there is a direct link between NFATc1 and these two genes. However, the biological relevance of our conclusion derives from the analyses carried out, even if on a limited number of genes. Indeed, among all the genes analysed, GATA2 is the only one that shows a noticeable increase in its expression (32.4 fold change) when NFATc1 is repressed, in the presence of RANKL, while all the specific osteoclast hallmarks (CtsK, MITF, Oscar, DC-STAMP, Acp5, MMP9) are definitely inhibited. Therefore, we conclude that GATA2 could contribute to a different pathway of macrophages differentiation, which does not lead to osteoclasts formation.

7. The manuscript has several grammar and formatting issues, including non-standard English (such as up- and down- expressed, instead of up regulated and down regulated), changes in line spacing, and seemingly random italicization.

7. As suggested by the reviewer we have checked the manuscript for mistakes.

Round  2

Reviewer 2 Report

The authors adequately addressed my concerns.